# Genetic Diversity and Genome-Wide Association Study for the Phenology Response of Winter Wheats of North America, Western Asia, and Europe

**DOI:** 10.3390/plants12234053

**Published:** 2023-12-01

**Authors:** Adil El Baouchi, Mohammed Ibriz, Susanne Dreisigacker, Marta S. Lopes, Miguel Sanchez Garcia

**Affiliations:** 1International Center for Agricultural Research in the Dry Areas (ICARDA), Rabat 10100, Morocco; 2Genetic and Biometric Laboratory, Faculty of Sciences, Ibn Tofail University, Kenitra BP. 242, Morocco; m_ibriz@yahoo.fr; 3The International Maize and Wheat Improvement Center (CIMMYT), Texcoco 56237, Mexico; s.dreisigacker@cgiar.org; 4The International Maize and Wheat Improvement Center (CIMMYT), Ankara 3906511, Turkey; marta.dasilva@irta.cat; 5Sustainable Field Crops, Institute for Food and Agricultural Research and Technology (IRTA), 251981 Lleida, Spain

**Keywords:** winter wheat panel, genetic diversity, population structure, vernalization, photoperiod, phenology, genome-wide association study

## Abstract

Wheat is a staple food in many areas around the World. In the 20th century, breeders and scientists were able to boost wheat yield considerably. However, a yield plateau has become a concern and is threatening food security. Investments in cutting-edge technologies, including genomics and precision phenology measurements, can provide valuable tools to drive crop improvement. The objectives of this study were to (i) investigate the genetic diversity in a set of winter wheat lines, (ii) characterize their phenological response under different vernalization and photoperiod conditions, and (iii) identify effective markers associated with the phenological traits. A total of 249 adapted genotypes of different geographical origin were genotyped using the 35K Axiom^®^ Wheat Breeder’s Array. A total of 11,476 SNPs were used for genetic analysis. The set showed an average polymorphism information content of 0.37 and a genetic diversity of 0.43. A population structure analysis revealed three distinct subpopulations mainly related to their geographical origin (Europe, North America, and Western Asia). The lines of CGIAR origin showed the largest diversity and the lowest genetic distance to all other subpopulations. The phenology of the set was studied under controlled conditions using four combinations of long (19 h light) and short photoperiod (13 h light) and long vernalization (49 days at 5 °C) and no vernalization. With this, phenological traits such as earliness per se (Eps), relative response to vernalization (RRV), and relative response to photoperiod (RRP) were calculated. The phenotypic variation of growing degree days was significant in all phenology combinations. RRV ranged from 0 to 0.56, while RRP was higher with an overall average of 0.25. The GWAS analysis detected 30 marker-trait associations linked to five phenological traits. The highest significant marker was detected on chromosome 2D with a value of −log10(*p*) = 11.69. Only four loci known to regulate flowering exceeded the Bonferroni correction threshold of −log10(*p*) > 5.1. These results outline a solid foundation to address global food security and offer tremendous opportunities for advancing crop improvement strategies.

## 1. Introduction

Bread wheat (*Triticum aestivum* L.) is a vital crop for food security with a global production of 760 M tons [1] annually. Wheat also occupies the largest land under cultivation with more than 200 million ha planted and is considered as one of humanity’s key sources of calories, contributing to up to 20% of its daily intake worldwide. To fulfill the demand for human food consumption by 2050, global wheat production need to increase by almost 70% [2,3]. Several studies show the importance of genetic diversity for breeding programs to be successful and highlight the importance of adaptive genes in the improvement of wheat [4,5,6,7,8,9]. Given the impact of climate change, breeding for earlier or later maturing lines as well as optimizing other phenology traits becomes crucial to ensure sustainable wheat adaptability and production. Continuous selection for genes or ideotypes can lead to the decrease and drift in allele frequency and therefore to a loss of genetic diversity. In breeding populations, genomic region consistently correlated with important traits reflect major drivers of genome divergence [10]. Ref. [11] noted that in modern wheat, a restricted number of loci reflect the phenotypic improvement, while the largest proportion of the genome remains unchanged. Vernalization, photoperiod sensitivity, and earliness per se are regulated by *Vrn*, *Ppd*, and *Eps* genes, respectively, and influence the capacity of wheat to adapt to a particular cropping region. All genes are key regulators of the shift from the vegetative stage to the reproductive development when the temperature is warm enough and rainfall is sufficient. Insights in the genetic variation for these genes (*Vrn*, *Ppd*, and *Eps*) in winter wheat provides breeding programs with a source of alleles to improve adaptation given the current requirement for adaptability to changing weather patterns. 

Vernalization is a crucial process protecting the floral meristems from frost damage [12]. The process is controlled by four homoeologous loci: *Vrn-1* [13], *Vrn-2* [14], *Vrn-3* [15], and *Vrn-4* [16]. Winter wheat varieties carry recessive vernalization alleles [17]. After the vernalization requirement is met, the photoperiod response, which is primarily influenced by the *Ppd-1* loci, governs and adjusts flowering timing for specific environments [18]. Earliness per se or “narrow-sense earliness” is a quantitative trait detected when vernalization and photoperiod requirements are satisfied [19]. The mayor genes at the *Vrn* and *Ppd* loci have been cloned and the effect of diverse alleles studied. Gene-specific markers established are commonly used in wheat breeding programs [20]. Conducting phenological studies under diverse controlled growth conditions presents an alternative for characterizing adaptability, particularly when simulating field conditions [21]. 

Genome-wide association studies (GWAS) have become an extremely popular and cost-effective approach for the identification of molecular marker related to complex traits of interest [22,23,24,25,26]. The further understanding of phenological traits under controlled conditions and the identification of additional molecular markers are beneficial for future wheat improvement. GWAS using winter wheat germplasm has already been employed to examine the genetic bases of several traits [27,28,29]. GWAS can also be used to validate and explore allele effects of known candidate genes [30]. The primary goal of this research was to (i) explore the genetic diversity and population structure of the winter wheat panel WWAGI, (ii) examine the phenological response of the panel under different controlled vernalization and photoperiod treatments, and (iii) validate and identify molecular markers associated with phenological traits.

## 2. Results

### 2.1. Genetic Diversity, Relationship, and Population Differentiation

A total of 11,476 polymorphic SNPs were used for genetic analyzes. The B genome held most polymorphic SNPs (50%), followed by the A (40%) and D genome (10%). The chromosome with most SNPs was chromosome 5B (815), while the lowest number of SNPs was noted for chromosomes 4D (32). The average value of genetic diversity (GD) was 0.43 and ranged from 0.38 (chromosome 2A) to 0.47 (chromosome 3B) (Table 1).

The PIC values ranged from 0.33 of chromosome 2A to 0.4 of 3B, with an average of 0.37. Lines from Western Asia exhibited the highest GD with a value of 0.42, followed by lines originating from the CGIAR and lines from North America with a value of 0.39. The lowest GD was observed within lines from Central Asia, which amounted to 0.27. Regarding PIC, the same pattern was identified, ranging from 0.22 for lines from Central Asia to 0.37 for lines derived from Western Asia with an average of 0.32. For the whole panel, the mean LD (R2) decreased as genome distance increased, indicating that the probability of LD was low across widely separated SNP pairs with a rate of 1 Mbp (Appendix A).

The PCA revealed sparse grouping according to geographic and institutional origin. The first axis explained 9.94% of the total genotypic variance, the second axis 3.12%. The first axis differentiated between wheat from Central Asia and European lines, which are located on opposite sides of the axis. The second axis separated between the North American and the Western Asia wheat, positioning them on opposite sides of the axis (Figure 1A). 

In addition, Western Asian lines were separated across the second axis, with Iranian lines predominantly on one side and those near European lines, principally from Turkey, on the other side. In agreement with the PCA, AMOVA results indicated that the variance within subpopulations explained most of the genetic variation (94%; *p* ˂ 0001), whereas variation among subpopulations was only 6% (*p* ˂ 0001) (Table 2). AMOVA results within groups showed that the highest proportion of genetic variance (26%) was observed within lines from the CGIAR (IWWIP, CIMMYT, and ICARDA) followed by the subpopulation of North America and Western Asia with 23% and 22%, respectively. However, the Western Asian subpopulation (mostly from Turkey and Iran) showed the largest genetic variation per entry as indicated by the mean squares (Table 2). 

To determine the genetic diversity trend across geographic origin, the genetic distance among the six subpopulations was compared. The Nei’s genetic distance ranged from 0.012 to 0.1 with an overall mean distance of 0.05. The CGIAR subpopulation showed the lowest genetic distance with all other subpopulations with an average of 0.037 (Table 3). 

### 2.2. Characterization Based on Gene Diversity for Ppd and Vrn

Regarding the major flowering time genes, 19% of the genotypes carried *Ppd-B1a* and 80% *Ppd-B1b*, while 61% carried *Ppd-D1a* and 31% the sensitive *Ppd-D1b* allele (Figure 2B). The *Ppd-B1b*/*Ppd-D1a* allelic combination was the most common in the WWAGI with 59% of the genotypes, followed by 22% for *Ppd-B1b*/*Ppd-D1b*. For the main vernalization genes, 89%, 76%, and 87% of the genotypes carried the winter type allele for *Vrn-A1*, *Vrn-B1* and *Vrn-D1*, respectively, while only 8% carried the spring allele for *Vrn-A1*, 21% for *Vrn-B1*, and 9% for *Vrn-D1* (Figure 2C). As expected, the triple winter *Vrn* alleles were the major component of the WWAGI with 66%, while all the other allelic combinations represented less than 16%. In addition, the AMOVA results suggest that the different allelic combinations of *Vrn* and *Ppd* had a limited effect on the population structure (Table 2). 

### 2.3. Population Structure

The population structure of the WWAGI was examined using the sNMF function, and the results showed that the cross-entropy criteria did not reveal a minimum value or distinct plateau and continuously dropped as k increased. However, significant results were observed at K2 and K3. At each K level, genotypes were allocated to different subpopulations, and genotypes with an ancestry coefficient less than 70% were classified as admixed and mapped to principal component analysis (PCA) clusters. At the K = 2 level, 20% were assigned to subpopulation 1 and 26% to subpopulation 2, while the remaining 54% were admixture. Most of subpopulation 1 comprised lines from Europe and Western Asia (64% and 22%). Subpopulation 2 contained lines from three groups, Western Asia, CGIAR, and North America, representing 91%. At K = 3, 64% of the lines were admixed, while the 36% remaining were divided between three subpopulations with 17%, 10%, and 8% for subpopulations 1, 2, and 3, respectively. Subpopulation 1 was composed mainly of the European lines, while subpopulation 2 was largely made of North American lines. For subpopopulation3, 57% were representing Western Asia and 33% represented lines from CGIAR (Figure 1C).

In the geographical region, *Vrn-1* and *Ppd-1* genes are lined up against the phylogenetic trees (in three different layers). The phylogenetic can be interpreted to split the dataset into four clades (Figure 2A). The subgroup I was mainly composed of genotypes coming from North America with few lines from Europe and CGIAR, and the second subgroup II principally combines Lines from Europe with few from Central, Western Asia, and CGIAR, while subgroup III comprises genotypes coming from North America, CGIAR, and Europe. The fourth subgroup is primarily composed of lines from Western Asia and CGIAR. The first and the second subgroup (I and II) englobed essentially triple winter types for the three copies of the *Vrn* gene (Figure 2A). For the *Ppd* gene, the first subgroup is mainly composed of the double sensitive alleles at *Ppd-B1b* and *Ppd-D1b*, and the sensitive *Ppd-B1b* and insensitive *Ppd-D1a*. At the same time, the second subgroup is predominantly made of the sensitive/insensitive *Ppd-B1b*/*Ppd-D1a*. For the subgroup III, the *Vrn* gene is represented by the triple winter type and double recessive winter type, while for the *Ppd* gene, the subgroup III englobed the sensitive/insensitive *Ppd-B1b-*/*Ppd-D1a* mainly, followed by the insensitive/sensitive *Ppd-B1a*/*Ppd-D1b*. The last subgroup IV was represented mainly by the double winter type and double recessive winter type for the *Vrn* gene, and by the sensitive/insensitive *Ppd-B1b*/*Ppd-D1a* for the *Ppd* gene.

### 2.4. Response to Photoperiod and Vernalization

The results of the phenology experiment (Table 4) showed that the genotype effect was significant in all treatments for the GDD to heading date. GDD ranged from 1013.9 to 1629.42 with an average of 1273.13 for full pre-vernalization and a long photoperiod (V+P+), whereas for no vernalization and a long photoperiod (V−P+), it ranged from 896.85 to 2241.87 with an average of 1417.13. The highest average was found for V+P− with 1705.05, which ranged from 1270 to 2150. Across all treatments, the GDD showed high heritabilities (Table 5).

The highest heritability was detected for V+P+ (h = 0.92) followed by V−P+ with 0.91 and V+P− by 0.81. On the other hand, there was a strong significant difference for GDD across treatments and the interaction of genotype × treatment (Table 6).

For the response to vernalization and photoperiod treatments of the WWAGI, the experiment revealed that among all lines, only 35% (85 lines) were able to flower under no vernalization, whereas 70% of the lines (179) under short photoperiod were able to flower. The results showed that the presence of one spring allele is sufficient to trigger flowering for 78% of the lines carrying one (Figure 3A). However, just 36% for Vrn-A1, 27% for Vrn-B1, and 36% for Vrn-D1 with winter alleles were able to flower. At the same time, the percentage of flowering was almost similar for the lines carrying one or two spring alleles at the *Vrn* loci with 83% and 85, respectively, while it was only 15% for the triple winter type (Figure 3B). Most of the winter wheat lines that flowered without vernalization still showed a quantitative response to vernalization, and this response was associated with the *Vrn* allele they carried. Thus, for the average extra days to flowering under no vernalization, the lines carrying the three winter alleles showed the highest value with 33 extra days as compared to the vernalized treatment. The lines carrying winter alleles at *Vrn-B1* showed a delay in flowering under no vernalization of 35 days, while the genotypes with spring alleles revealed a lower number of extra days to flowering with only 9 extra days for *Vrn-A1* and 22 days for *Vrn-B1* as compared to growing under 49 days of vernalization. At the same time, *Vrn-D1* did not show a clear effect as compared to its spring version (Figure 3C). This result indicated that the *Vrn-A1* gene contributed the most to flowering time under non-vernalization. On the other hand, *Vrn-D1* showed the weakest effect on flowering, and therefore, the presence of its spring or winter allele did not delay the flowering time significantly. Furthermore, considering any winter allele, there was a noticeable increase in the average extra days to flowering when comparing genotypes with a single winter allele to those with double or triple winter alleles. Genotypes with a single winter allele exhibit a delay of 14 days, while it was 22 and 49 days for double and triple winter allele genotypes, respectively (Figure 3D). The number of winter alleles increased the extra days to flowering by 71% from single to triple. Regarding the photoperiod treatment (Figure 3E), the percentage of lines that achieved flowering under short photoperiod was 100% for lines with both insensitive *Ppd* alleles, while the percentage was reduced by only 9% for those carrying the sensitive *Ppd-B1b*. However, for those carrying the sensitive *Ppd-D1b* allele, the percentage of flowering was decreased to 56%, while it was only 11% for the lines carrying both sensitive *Ppd* alleles (*Ppd-B1b*/*Ppd-D1b*). Most lines that flowered under a short photoperiod also showed a quantitative response to a long photoperiod, which was related to the *Ppd* allele they carried (Figure 3E). In addition, the highest value of extra days under short photoperiod was detected for the *Ppd-B1b*/*Ppd-D1b* combination with 21 days alongside with *Ppd-B1a*/*Ppd-D1b* with 22 days, while *Ppd-B1b*/*Ppd-D1a* and *Ppd-B1a*/*Ppd-D1a* combinations showed a lower number of days with 15 and 17 days, respectively (Figure 3F). This suggested that *Ppd-D1* is the strongest regulator of flowering time under a short photoperiod, while no significant effect of *Ppd-B1* on flowering was detected.

The phenology experiment revealed that out of the 249 lines used, only 31 lines needed both vernalization and a long photoperiod to flower, while 102 lines required only vernalization, 20 lines needed a long photoperiod, and 65 genotypes required neither (Figure 4).

In terms of region-specific distribution (Figure 5), the CGIAR lines were mostly composed of two major groups: lines that need vernalization but not a long photoperiod (49%), and those that did not require vernalization but required a long photoperiod (44%). This was similar for the North America lines with 45% and 40%, respectively. In the European group, most lines tested required vernalization but not a long photoperiod (76% of the lines), while 13% required a long photoperiod without vernalization. For Central Asia, three groups with identical percentages (33% each) were present. However, almost 88% of the lines coming from the Western Asia region either did not require vernalization and a long photoperiod, or they needed vernalization but not a long photoperiod (Figure 5). 

The quantitative response to the treatments allowed the calculation of the relative response to vernalization (RRV) and relative response to photoperiod (RRP) indices, quantifying the sensitivity of the WWAGI set to these. Across the WWAGI, and in absolute terms, the RRV ranged from 0 to 0.56 with an average of 0.2, while RRP was higher with an overall average of 0.25, but less variable. In comparison to Eps, both RRP and RRV showed a significant moderate correlation with R^2^ of 0.33 for RRP and of 0.12 for RRV (Figure 6A,B). However, no significant correlation was detected between RRP and RRV themselves (Figure 6C).

Furthermore, the results revealed that Eps was significantly higher for the lines that did not require vernalization to flower than for those who did. No difference could be found in Eps between the lines requiring or not a long photoperiod to flower. The genotypes that required vernalization to flower were mainly originated from Europe, North America, and CGIAR, whereas those from Central Asia and Western Asia (Iranian group) did not, in general, require vernalization (Figure 7E). Furthermore, 77% were also photoperiod-insensitive, while the 23% remaining were photoperiod-sensitive. In addition, most North America lines and 6% from the CGIAR group exhibited photoperiod sensitivity, whereas the remaining were found to be insensitive to the photoperiod. This suggests that there may be a potential relationship between geographic origin and photoperiod sensitivity in these lines (Figure 7F). Regarding the vernalization and photoperiod experiment (Figure 7G), PCA revealed two groups clustered with their respective vernalization and photoperiod requirements. Furthermore, no clear pattern was detected in the relationship between RRP, RRV, Eps, and population structure. However, a significant differentiation was apparent in the case of strict vernalization and photosensitivity. Specifically, most of the European and North American lines exhibited a strict vernalization requirement, while lines originating from Asia and CGIAR did not require vernalization. Regarding strict photosensitivity, only a subset of lines from North America and CGIAR required a long photoperiod, indicating that the population structure may be influenced by strict vernalization and photoperiod requirements.

### 2.5. Genome-Wide Association Study

GWAS for the traits studied identified 17 significant markers (6 in genomes A and B and 5 in genome D; −log10(*p*) > 3) located on chromosomes 1A, 2D, 5A, 5B, 5D, 7B, and 7D. The percentage of phenotypic variation explained by significant markers ranged from 9% to 49%. The highest percentage was observed for RRV with an average of 43%, while the lowest value was detected for RRP. The percentage of phenotypic variation explained by functional genes ranged from 12% to 44%, while from 9% to 45%, it was explained by the newly identified markers. Four significant markers were detected for strict photosensitive, distributed across three chromosomes and in charge of 25–41% of the phenotypic variation (Table 7).

The *Ppd-D1* in chromosome 2D was the most significant association (LOD = 11.8) and responsible for the highest proportion (41%) of the phenotypic variation for strict photosensitivity. The lowest number of MTAs were identified for Eps with only 1 marker *Vrn-B1* in chromosome 5B (LOD = 3.7). For strict vernalization, five significant markers were identified, distributed in chromosomes 1A, 5A, and 5B. The mean of the phenotypic variation explained by these markers was 29%. Allele *Vrn-B1* explained the highest phenotypic variation with 36% and was present in 54 lines; among them, 78% were characterized by strict vernalization, and all of them carry the winter type allele of *Vrn-B1*. Out of the seven MTAs, the remaining five were identified for RRV and two for RRP. Associated SNPs identified for each trait are listed with their respective LOD in Table 7. Only one marker was associated with two different traits: *Vrn-B1*, with strict vernalization and Eps. The marker AX-94796479 was present in only 17 lines, of which 11 showed no relative response to vernalization, and the remaining showed a response below 0.13 (Figure 8).

## 3. Discussion

### 3.1. Genetic Diversity and Population Structure

Wheat breeding is a continuous process and exploitation of genetic diversity is crucial. For attaining specific breeding goals not only is the discovery of genes important, but also the contribution of these genes needs to be analyzed and dissected, and their association with specific traits of interest needs to be assessed and quantified. This process of gene discovery will ultimately contribute to a deeper knowledge of the wheat genome and its function. Within this framework, we investigated the genetic diversity of a worldwide winter wheat association genetics panel, originated from seven different regions and 32 countries. This population showed high GD (0.47) and PIC (0.33), suggesting a higher levels of polymorphism as compared to previous studies [31,32,33]. Moreover, the B genome showed the highest GD and PIC followed by the A genome and finally the D genome, similarly to previous studies [33,34]. 

The results from the two clustering methods (PCA, and NJ tree) demonstrated the diversity of the 249 WWAGI lines grouped into three subpopulations (Europe, North America, and Western Asia) and the presence of additional subgroups characterized by common related origins (CGIAR). This can be attributed to the CGIAR’s utilization of the entire spectrum of genetic diversity available among modern varieties and the increased adoption of CGIAR advanced lines through worldwide distribution of international nurseries. Similarly, Ref. [35] studied a panel of 1000 wheat accessions from 91 countries and found that the population was clustered based on geographical origin in three main groups, European, Asian, and American. Geographic origin has previously been proven to be a major driver of population clustering. For instance, Ref. [36] showed in a panel of 3200 wheat accessions that the population was structured based on geographic origin and growth habit. Also, Ref. [37] revealed that using 635 wheat accessions that the geographic origin and growth habit were the primary basis for clustering. AMOVA analysis based on geographical origin revealed that more than 6% of the overall genetic variations were identified among the origin groups. These observations are comparable to those reported by Ref. [38] with a collection of 290 winter wheat varieties and by Ref. [39] with a panel of 753 U.S. wheat varieties including 517 winter wheat varieties, and are lower than those reported by Ref. [40], where 11% of variations were captured among countries. The modest diversity among the modern winter wheat panel (6% based on regions) displayed in Table 2 was most likely because of the large and regular sharing of parents among the wheat community. Nevertheless, these findings indicate that there is genetic diversity available for use within wheat breeding programs. The clustering patterns shown by PCA and NJ tree offer strong evidence that the panel contains a pool of unused genetic potential. By confirming the presence of distinct subpopulations, such as those associated with CGIAR lines, consequently, it is possible that these lines have valuable traits that can be used to enhance the wheat breeding program effort.

A large number of studies have used *Vrn* and *Ppd* alleles to explain genetic diversity and population structure [12,41,42]. In our study, we explored population structure taking into consideration the gene-specific molecular markers *Vrn* and *Ppd*. For the vernalization genes, *Vrn-A1a* was almost absent in the WWAGI with only 21 lines, while *vrn-A1* was present in 222 lines. Lines from CGIAR, Europe, and Western Asia lines have been characterized with photoperiod sensitivity at *Ppd-B1* and insensitive *Ppd-D1* as the most common allelic combination for *Ppd*, while the most common Central Asia was the double sensitive and the insensitive/sensitive *Ppd-B1a*/*ppd-D1b* for North America. However, the Central Asian genotypes was notably the smallest among the studied groups, which may raise a statistical limitation or potential bias due to the limited sample size. These outcomes are comparable to those discussed by [43] with 257 wheat cultivars from CIMMYT where 91% carried *Ppd-D1a*. However, Ref. [44] described that out of 245 European cultivars, 224 carried the sensitive allele *Ppd-D1b* and only 8.5% were carried the photoperiod-insensitive allele *Ppd-D1a,* found mainly in southern Europe, while Ref. [45] showed that 58% out of 521 wheat cultivars from Europe carried the insensitive allele *Ppd-D1a*. 

For *Vrn*, and as expected, the most common allelic combination was the winter type for all regions except Central Asian lines that were double winter type for *Vrn-A1* and *Vrn-B1* and mostly carried the spring type for *Vrn-D1*.

### 3.2. Phenology Variability and Marker Trait Association

Flowering time in wheat has previously been linked to variation in the *Vrn* and *Ppd* genes such as insertions, deletions, and mutations [46]. In this study, the panel was characterized in a growth chamber with controlled light and temperature, simulating different and extreme vernalization and photoperiod conditions, to determine the phenology response. In general terms, it has been established that spring wheat growth habit does not require any cold treatment to transition to flowering [47]. However, for winter wheat growth habit, as vernalization is a quantitative response, increasing the duration of vernalization until requirements are met result in accelerated flowering [48]. Under no vernalization, we observed that the percentage of flowering was more than 78% for lines carrying spring alleles at the *Vrn* loci, while it was less than 36% for those carrying winter *Vrn* alleles. All the lines that flowered under no vernalization still revealed a response to low-temperature exposure, and this response was linked to the allelic combination at *Vrn* loci. Specifically, lines carrying triple winter *Vrn* alleles had the lowest percentage of flowering (less than 15%) as compared to double and single *Vrn* type (more than 85%), reflecting the cumulative influence of *Vrn* genes on flowering. This highlights that *Vrn* genes impact significantly flowering as described by large body of research [48,49,50]. As expected, this allelic variation significantly impacted the extra days to flower, for *Vrn-A1* and *Vrn-B1* we found that the genotypes carrying the spring alleles showed the lowest average extra days to flowering, as compared to their response under long vernalization, from less than 22 days to more than 33 days for winter lines. Furthermore, we found that the difference in extra days to flowering between winter and spring alleles is higher at *Vrn-A1* with 24 days as compared to 13 days for *Vrn-B1*. This is in agreement with several research suggesting that *Vrn-A1* is the strongest driver of vernalization requirement [14,51,52], while no difference was detected between lines carrying the *Vrn-D1* spring allele and those carrying the winter allele. This finding is in agreement with results published by [53]. However, lines carrying the triple winter combination can flower under no vernalization, derived mainly from the CGIAR and Western Asia regions. This is probably due to another source of variation in vernalization response via copy number variation (CNV) as reported by [48,50] and/or additional *Vrn* genes, e.g., *Vrn-3*. CNV for *Vrn-A1* was reported as the highest among the *Vrn* loci [50,54], which can be from one to four copies and therefore may exhibit multiple effects on flowering, such that the higher the number of copies, the longer the vernalization duration required [55]. This suggests that the use of genetic markers associated with a *Vrn* allele in breeding programs may be enough to select for gross phenological adaptation, but for fine-tuning it, markers for CNV, either using associated haplotypes or through RT-PCR, are necessary.

Wheat photoperiodism is primarily regulated by the *Ppd-D1* locus on chromosome 2D, which strongly controls flowering time by separating lines into photoperiod-sensitive and -insensitive. As a result, it influences the proportion of lines that flower under a short photoperiod and therefore the extra days required for flowering [56,57]. Analogously, lines carrying the insensitive allele of *Ppd-D1a* showed the highest percentage of flowering under short photoperiod conditions with more than 91%, as compared to less than 56% for lines carrying the sensitive allele of *Ppd-D1*. Moreover, the average extra days under short photoperiod was lower for the *Ppd-D1*-insensitive allele (less than 17 days), while it was more than 3 weeks for the sensitive *Ppd-D1* allele. In addition, no difference was detected in the extra days to flowering between lines carrying the sensitive and the insensitive *Ppd-B1* allele but still impacting the absolute sensitivity to photoperiod. We found that lines carrying the sensitive allele at *Ppd-D1* exhibit a delay in flowering by 5.5 days. Similarly, Ref. [58] reported in a panel of 1100 winter wheat cultivars that the strongest association with phenology including heading date was observed for *Ppd-D1*, thus delaying flowering by 4.8 days [58]. It was demonstrated in a large panel of winter wheat that *Ppd-D1* was the primary switch to flowering and that the copy number at *Ppd-B1* was the second most important source of variation, although [59] reported that *Ppd-B1* could be as strong as *Ppd-D1*, and both of them significantly accelerate spike development under a short photoperiod. This is in agreement with previous observations [50,60,61]. Our results on the photoperiod response demonstrated that the sensitive allelic combination of both *Ppd-B1* and *Ppd-D1* plays a significant role in blocking flowering under a short photoperiod.

The geographic pattern distribution of the vernalization and photoperiod response experiment in our WWAGI revealed that most of the European lines (76%) required long vernalization to flower but could still do it under short days. Most probably, this is because most of them carry the insensitive *Ppd-D1* allele. Refs. [56,62] reported that southern and eastern European wheats were usually photoperiod-insensitive, while wheat from the northern latitudes of Europe were photoperiod-sensitive. Analogously, the majority of our European lines (90%) originated from eastern and southern Europe. Regarding North American lines, the experiment demonstrated that the vast majority (85%) need long vernalization to flower, and 40% of these lines require long photoperiods as well. These results concur with those reported by [63] in Canadian winter wheat material with a diversity panel of 203 winter wheat genotypes and [64] using 299 winter wheat genotypes from USA. Almost all the CGIAR lines (93%) were photoperiod-insensitive with 49% of them requiring long vernalization, suggesting that all these lines were adapted to lower latitudes. This is expected given that the vast majority of CGIAR wheat have insensitive photoperiod genes in their ancestors [49]. The Central Asia lines showed the highest proportion of photoperiod-sensitive lines (66%), and this is probably due to the high latitude at which they are grown. Nevertheless, it is important to mention that these genotypes represented the smallest group among the studied groups, which may introduce potential statistical limitations. The distribution across the Western Asian lines was similar to CGIAR lines with slight differences in those that do not require vernalization and long photoperiod. The proportion of the Western Asian lines that did not require vernalization and long photoperiod was 8% higher than the CGIAR lines, while they were less with 14% for genotypes that needed vernalization but not a photoperiod. Strict vernalization and photoperiod requirements exhibited a major impact on population structure since the European and North American Lines were clustered together as strict vernalization, while Asian and CGIAR lines were not. Similarly for strict photosensitivity, one cluster from North America and CGIAR were sensitive to a long photoperiod, confirming that selection based on vernalization and photoperiod requirements play an important role in population structure. 

To identify unique and novel MTAs for specific traits and under specific conditions, genome-wide association studies have been used extensively in wheat [23,25,65,66,67,68]. Previous studies have revealed that genes involved in flowering are mostly found on chromosomes 1A, 2B, 3A, 3B, 5A, 6A, 6B, 7A, 7B, and 7D [69,70,71]. In the current study, a total of 30 significant markers associated with five phenological traits were identified. The high significant functional marker *Ppd-D1* was found for the strict photosensitive which is consistent with previous genetic research that identified the *Ppd-D1* gene is the most efficient photoperiod response gene, followed by *Ppd-B1* and *Ppd-A1* [23,60]. For the strict vernalization, we identified two significant MTAs of *Vrn-B1* located on chromosome 5B; these identified genes serve as the primary regulators of plant growth in the vernalization pathway [13,72], and are overlapping with several studies previously reported [12,16,41,45,73]. The *Vrn-B1* gene involved considerably in the regulation of flowering, and it has been found to be impacted by copy number variation [48,50]. During the investigation, we found that lines with the *Vrn-B1a* allele flowered 12 days earlier than those with *Vrn-B1*. This finding is in good agreement with the results published by [23], which revealed the same earliness of flowering for cultivars with allele *Vrn-B1a* but with just 3 days. We have observed that the LOD score of *Ppd* was higher than *Vrn*, and our result demonstrated that *Ppd-D1* gene had the highest effect on flowering followed by *Vrn-B1*. The authors of [23] revealed that *Ppd-D1* had the most important effect on heading and flowering then *Vrn-B1* in Chinese wheat. This is due to the fact that *Vrn* genes are positioned in a more dynamic region of the wheat genome as compared to *Ppd* genes; as a result, the *Vrn* genes have more allelic variation and genetic diversity than *Ppd* genes. Furthermore, it is demonstrated that *Eps* genes are located near the centromere of 5B chromosome [74]. Until now, *Eps-A^m^1* in Triticum monococcum on chromosome 1A is the best identified *Eps* locus in wheat [19,75]. Our results indicated that gene *Vrn-B1* on chromosome 5B which is related to vernalization, has a significant impact on *Eps* and therefore on the flowering time in wheat. 

## 4. Materials and Methods

### 4.1. Plant Material

For this study, a panel of 249 advanced lines and varieties from the Winter Wheat Association Genetics Initiative (WWAGI) developed by the International Maize and Wheat Improvement Center (CIMMYT) was used (Appendix A). The lines originated from five different regions—Central and Western Asia, Europe, and North America—or originated from the CGIAR International Winter Wheat Improvement Program (IWWYP) a joint venture of the Turkish Government, the International Center for Maize and Wheat improvement (CIMMYT), and the International Center for Agricultural Research in the Dry Areas (ICARDA). The genetic material used are representative of the International Winter Wheat Improvement Program germplasm dissemination work (International Nurseries) and its international collaboration. Constituting a set of relevant breeding material representative of a diverse set of winter wheat breeding programs. Only a smaller number of lines derived from other regions. The largest number of lines (65) originated from the CGIAR (CGR), followed by United States (61) and Iran (27), Turkey (22), Russia (10), Ukraine (9), Bulgaria (6), and Moldova (5) (Figure 9A,B).

### 4.2. Genotyping

The WWAGI was genotyped with the 35K Axiom^®^ Wheat Breeder’s Array (Affymetrix UK Ltd., High Wycombe, UK) as per manufacturer’s guidelines [76]. The array comprised a total of 35,143 SNP markers. Genomic DNA was isolated from plants at the growth stage. Modified CTAB method was applied for the DNA isolation [77]. *Vrn-1* alleles (*Vrn-A1*, *Vrn-B1*, and *Vrn-B3*) were characterized to determine the growth habit of each genotype. The gene-specific markers described by Refs. [17,78] were used to identify variation in the promoter and intron-1 region of the *Vrn-1* genes. In addition, the presence of a SNP, located in Exon 4 of *Vrn-A1* was tested [52]. *Ppd-1* alleles which confer the sensitivity to day lengths were genotyped using gene-specific markers related to the junction between intact *Ppd-B1* copies in the ‘Sonora64’ allele and the 2089 bp deletion in the *Ppd-D1* gene [79]. 

Polymerase chain reaction (PCR) amplifications for STS markers were performed in a total volume of 10 μL containing a final concentration of 1× Buffer with Green Dye (Promega Corp. Madison, Wisconsin, USA), 200 μM deoxynucleotide triphosphates (dNTPs), 1.2 mM magnesium chloride (MgCl_2_), 0.25 μM of each primer, 1U of DNA polymerase (GoTaq^®^Flexi, Promega Corp., Cat. # M8295), and 50 ng of DNA template. PCR conditions were performed using the following temperature profile: 94 °C for 2 min followed by 30 cycles of 94 °C for 1 min, 54–60 °C for 2 min (dependent on the primer), and 72 °C for 2 min. PCR amplified products were separated by electrophoresis on 1.2% agarose gels using 1× TAE buffer, visualized under UV light. The SNP polymorphisms were scored using KASP reagents. Reactions contained 2.5 mL of water, 2.5 mL of 2× KASPar reaction mix, 0.07 mL of assay mix, and 50 ng of dried DNA with a PCR profile of 94 °C for 15 min activation time, followed by 20 cycles of 94 °C for 10 s, 57 °C for 5 s, and 72 °C for 10 s, followed by 18 cycles of 94 °C for 10 s, 57 °C for 20 s, and 72 °C for 40 s. Fluorescence was read as an end point reading at 25 °C. 

After removing markers with minor allele frequencies (MAF) of <5% and markers with more than 10% missing data, a refined set of 11,476 Axiom SNPs markers remained. The distribution of the filtered SNPs across the seven wheat chromosomes (Appendix A) was drawn using the Cmplot package for R 4.1.1 statistical software (R Core Team, 2019, R Foundation for Statistical Computing, Vienna, Austria. https://www.R-project.org/) [80]. 

### 4.3. Population Structure 

The Remington approach was used to conduct a linkage disequilibrium (LD) study [81]. The population structure was assessed using Neighbor joining (NJ) tree and principal component analysis (PCA). NJ analysis was conducted using the “BiocManager” and “ggtree” R packages [82], while LD and PCA were performed using TASSEL version 5.0 [83]. To further uncover population structure of WWAGI, a sparse “Non-negative Matrix Factorization” (sNMF) algorithm in LEA package was performed in R [84]. The method consists of estimation of admixture coefficient for each individual separately without taking into consideration a previous population membership supposition. The cross-entropy criterion from the sNMF function was used to check various potential population sets ranging from K = 2 to K = 10. Admixed lines were set to have less than 70% ancestral membership to any subpopulation.

### 4.4. Diversity Analysis

Major allele frequency (MAF), genetic diversity (GD), and polymorphism information content (PIC) for all chromosomes and individual regions and origin of winter wheat lines were calculated using PowerMarker version 3.25 software [85]. The analysis of molecular variance (AMOVA) and Nei’s genetic distance was performed using the same software to estimate genetic diversity between and within the predefined subpopulations. The optimal distances were assessed by a permutation test of 1000 [85]. 

### 4.5. Growth Chamber Experiment and Phenology Measurements

Three independent experiments were carried out in a climate-controlled growth chamber at 25 °C ± 1 °C temperature with a combination of two photoperiod (long photoperiod ‘P+’ and short photoperiod ‘P−’) and two vernalization durations (‘V+’ for full pre-vernalization and ‘V−’ for no vernalization). The details of the temperature and light treatments are provided in Table 4.

The experiments were set up growing one plant per genotype in a 1 L pot filled with peat moss and arranged in an augmented design with two repeated check cultivars NEKOTA and SERI, repeated 15 times per experiment. For each experiment both, days from germination to heading and growing degree days (GDD) for the same period were computed adopting the method of [72,86]. Extra days to flowering were assessed by deducting days required to flower under non-vernalization or under short photoperiod from days needed to flowering under vernalization or long photoperiod. Three phenological traits were calculated: earliness per se (Eps) was obtained using the rate of development (R) under long photoperiod and full vernalization, and relative response to vernalization (RRV) and relative response to photoperiod (RRP) were calculated according to [21,87] as follows:$$R=\frac{1}{F}\;\;\;\;\;\;\;\;\;RRV=1-\frac{RNV}{RFV}\;\;\;\;\;\;\;\;\;RRP=1-\frac{RSP}{RLP}$$

F = growing degree days from germination to heading date. RNV = rate of development under long photoperiod and no vernalization. RFV = rate of development under long photoperiod and full vernalization. RSP = rate of development under short photoperiod and full vernalization. RLP = rate of development under long photoperiod and full vernalization.

### 4.6. Statistical Analysis

For each experiment, a linear mixed model with spatial adjustment was built and used to determine adjusted means. The R package “statgenSTA” [88] was used to fit the model, which comprised a fixed effect for genotype and block, and an additional spatial component derived using Spatial Analysis of Field Trials with Splines (SpATS), which utilizes two-dimensional smoothing with P-splines as reported by [89]. Heritability (h2) was computed as a function of the effective dimension (ED) to the genetic component (EDg) h2 = EDg/(ng−l), where ng is the number of genotypes [90]. BLUEs (best linear unbiased estimators) were estimated with genotypes considered as fixed effect, heritability was treated as random [91].

### 4.7. Genome-Wide Association Study (GWAS)

The filtered SNPs, gene-specific marker results, and the BLUEs for each quantitative phenological trait were used for the GWAS analysis using GAPIT V3 (Genomic Association and Prediction Integrated Tool) in R [92]. Three models were performed: GLM [93], MLM [94], and CMLM [95] (https://github.com/jiabowang/GAPIT3 (accessed on 8 September 2023)). In addition to PCA, a relative kinship analysis was carried out to identify the genetic relationship between genotypes using the VanRadden method implemented in GAPIT [96,97]. Generalized linear mixed-model association test (GMMAT) was used for testing the vernalization requirement and photoperiod sensitivity (strict vernalization and photoperiod) [98]. Marker trait associations (MTA) were declared if their −log10(*p*) value was higher than 3.0. The significance threshold for MTA was elaborated based on Bonferroni correction of *p* ≤ 0.05/n, where n is the number of SNP markers used. Due to the conservative nature of the Bonferroni adjustment and to avoid false negatives, markers with an LOD > 3 are also highlighted and reported. A Q–Q plot with a predicted vs. observed log10(*p*) value was used to determine the quality of the fitted GWAS model. Manhattan plots was drawn using the CMplot R package (https://cran.r-project.org/web/packages/CMplot/index.html) to visualize the MTA.

## 5. Conclusions

Wheat is a crucial crop for global food security. In the present study, the WWAGI panel has been studied. The characterization and genetic dissection of the vernalization and photoperiod diversity of this panel added more value to our understanding of linkages, diversity, and population structure, while also making it easier to identify the best sources of genetic variability. As well as their relationship with phenology traits, quantitative and qualitative vernalization and photoperiod requirement in controlled condition can provide a valuable alternative to fields trials as it appears to avoid numerous constraints that may affect response accuracy. Additionally, this information will support the breeding program to improve the adaptation of the new winter wheat varieties. Genome-wide association for phenology in the WWAGI showed that a number of regulatory vernalization and photoperiod genes, such as *Ppd-D1*, *Vrn-A1*, and *Vrn-B1*, have a significant impact on the genetic architecture of flowering time. However, the distribution of significant MTAs led to the conclusion that there are many additional important genetic loci involved.

## Figures and Tables

**Figure 1 plants-12-04053-f001:**
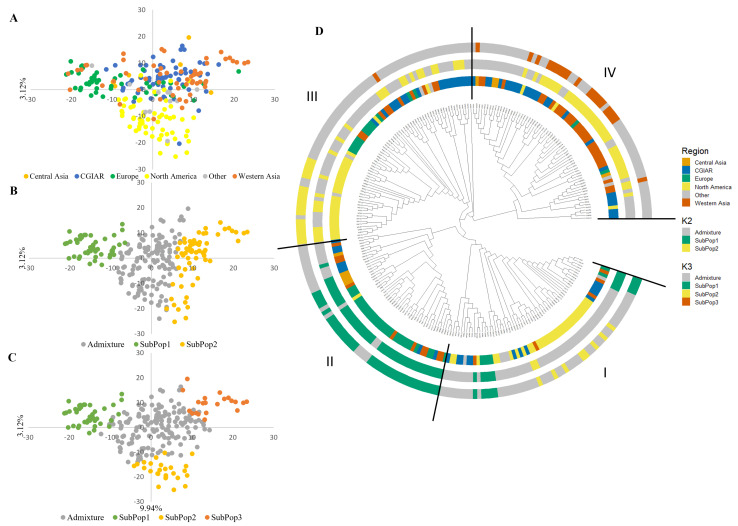
The genetic relationships and population structure of the Winter Wheat Association Genetics Initiative (WWAGI) with 249 advanced lines and varieties using principal component analysis (PCA) of SNP markers concluded via PCA and sparse Non-negative Matrix Factorization (sNMF) analyses. Spatial distribution of the genotypes is based on geographic origin (**A**) and assignment to sNMF groups at K = 2 (**B**), K = 3 (**C**). The phylogenetic tree of the WWAGI represents a clustering of individuals, through respective layers (from inside to outside), geographical region, and assignment to sNMF groups at K = 2 and K = 3. Genotypes with the highest ancestry coefficient below 70% are colored in gray (**D**).

**Figure 2 plants-12-04053-f002:**
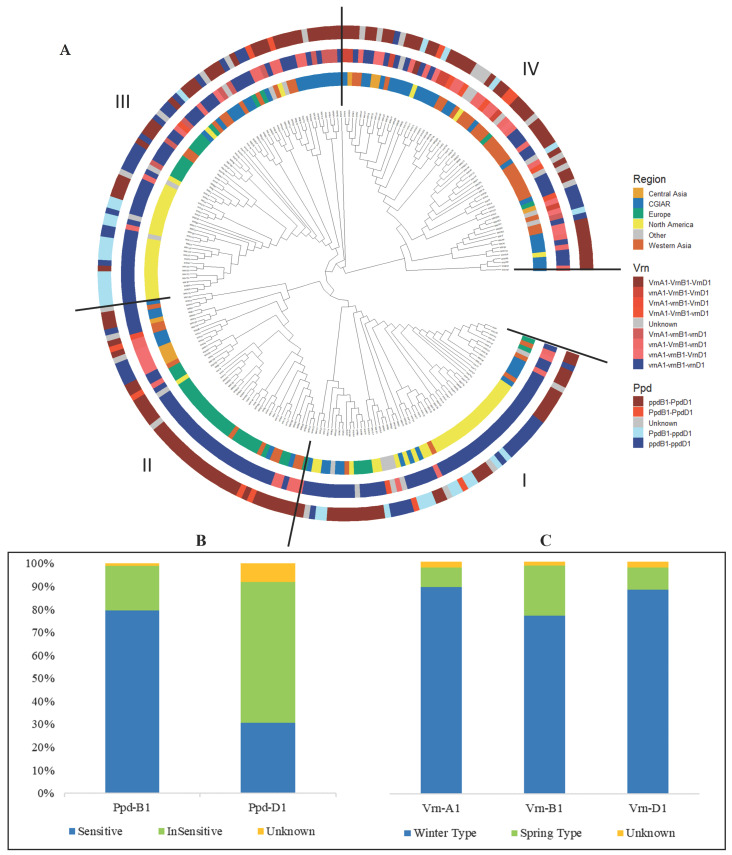
(**A**) Phylogeny-based clustering tree represents a clustering of individuals, through respective layers (from inside to outside), geographical region, and *Vrn* and *Ppd* allelic combination. (**B**,**C**) Characterization of the genotypes used based on gene diversity for *Ppd* and *Vrn*.

**Figure 3 plants-12-04053-f003:**
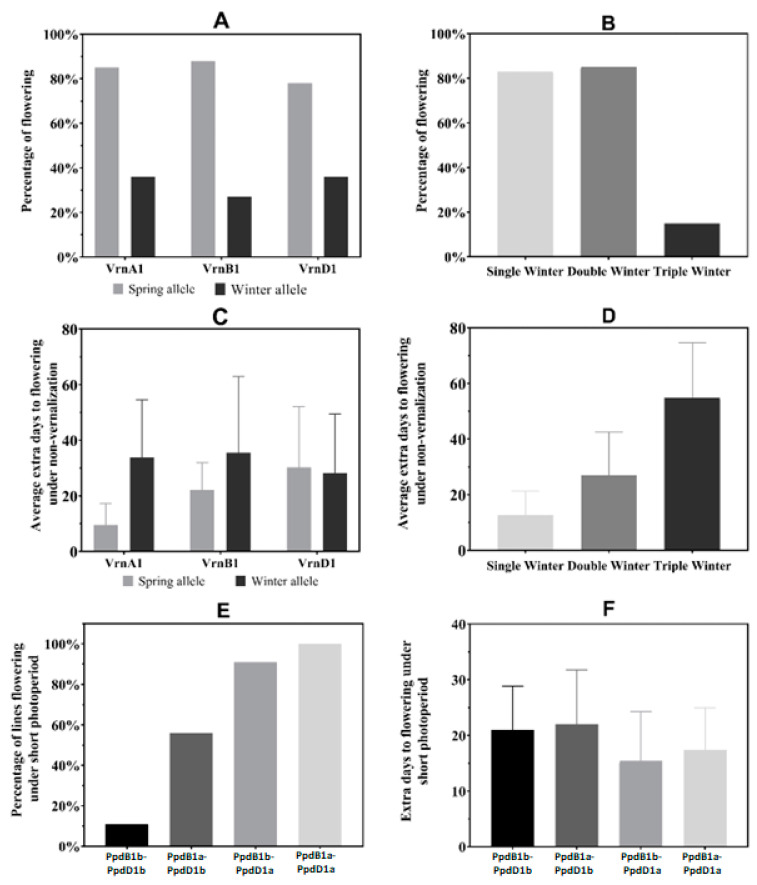
Unraveling regulatory flowering genes, vernalization response, and photoperiod sensitivity of the Winter Wheat Association Genetics Initiative (WWAGI). (**A**) Impact of *Vrn* Genes on percentage of flowering in spring and winter Alleles. (**B**) Impact of different winter types alleles on percentage of flowering. (**C**) Impact of *Vrn* Genes on average extra days to flowering in Spring and Winter Alleles under no vernalization. (**D**) Impact of different winter types alleles on average extra days to flowering under no vernalization. (**E**) Impact of *Ppd-B1*/*Ppd-D1* combination alleles on percentage of flowering under short photoperiod. (**F**) Impact of *Ppd-B1*/*Ppd-D1* combination alleles on average extra days to flowering under short photoperiod.

**Figure 4 plants-12-04053-f004:**
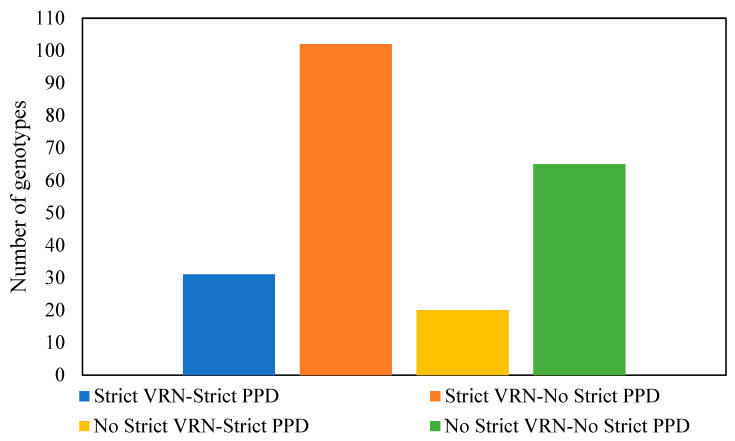
Phenology characterization of the winter wheat panel based on vernalization–photoperiod response requirement.

**Figure 5 plants-12-04053-f005:**
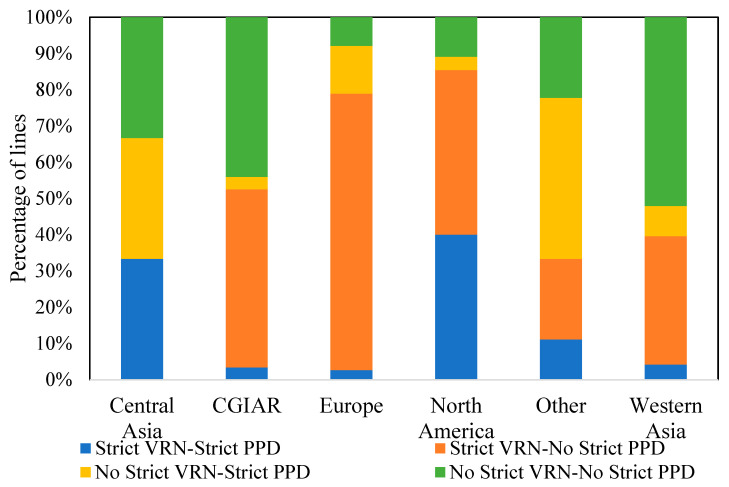
Phenology characterization of the winter wheat panel based on region and vernalization–photoperiod response requirement.

**Figure 6 plants-12-04053-f006:**
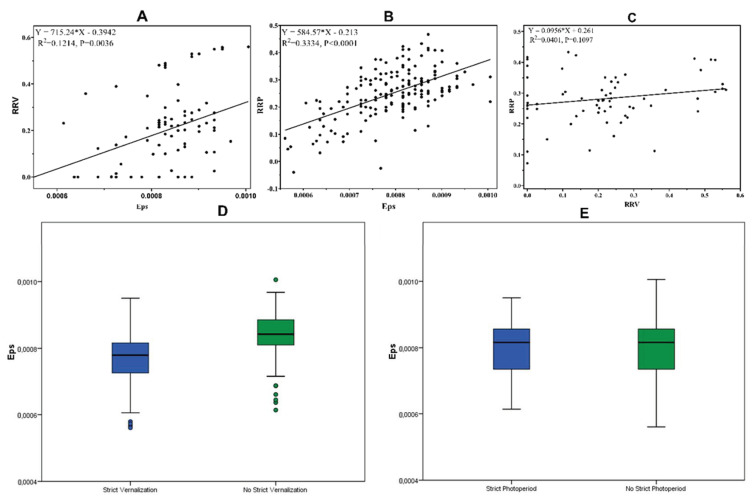
Relationship between relative response to vernalization (RRV), relative response to photoperiod (RRP), and earliness per se (Eps). (**A**) Regression between Eps and RRV. (**B**) Regression between Eps and RRP. (**C**) Regression between RRV and RRP. (**D**) Boxplot analysis of the Eps under vernalization treatment. (**E**) Boxplot analysis of the Eps under photoperiod treatment.

**Figure 7 plants-12-04053-f007:**
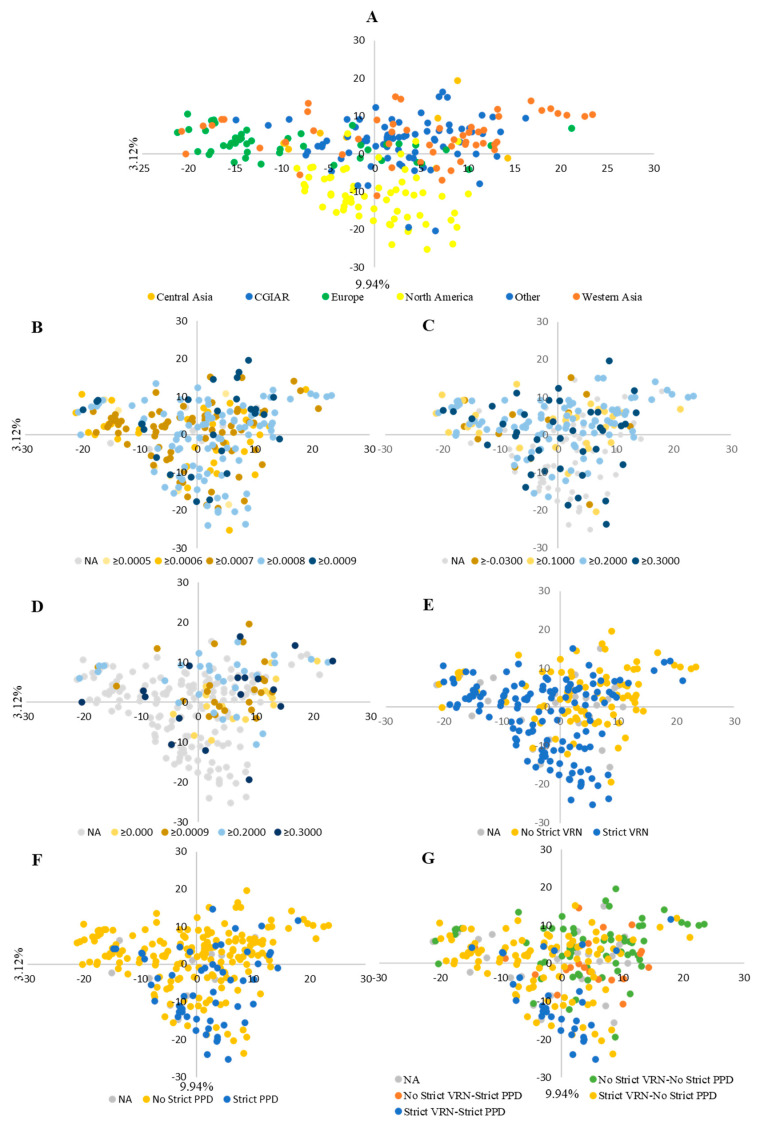
PCA based on geographic region (**A**), earliness per se (**B**), relative response to photoperiod (**C**), relative response to vernalization (**D**), complete vernalization (**E**), complete photosensitive (**F**), and vernalization and photoperiod requirement (**G**).

**Figure 8 plants-12-04053-f008:**
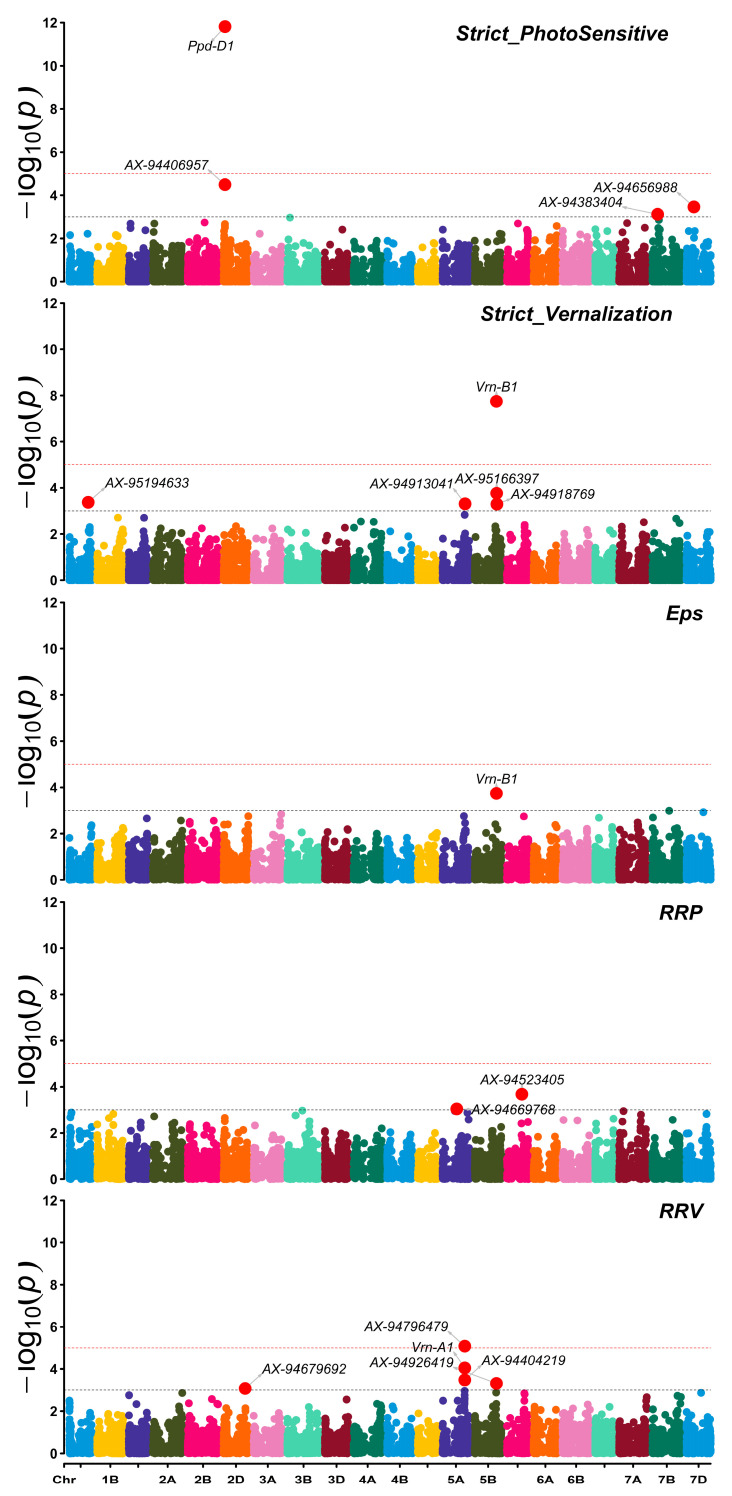
Genome-wide association scan for RRV, RRP, Eps, complete photosensitive, and complete winter in WWAGI. The plots show Manhattan plots with the names of the significant SNPs. The chromosomes are shown on the *X*-axis, and the genome-wide scan −log10 (*p*-values) values are shown on the *Y*-axis using common threshold (−log10*p* > 3) in black dotted line and Bonferroni correction (−log10*p* > 5.2) in red dotted line.

**Figure 9 plants-12-04053-f009:**
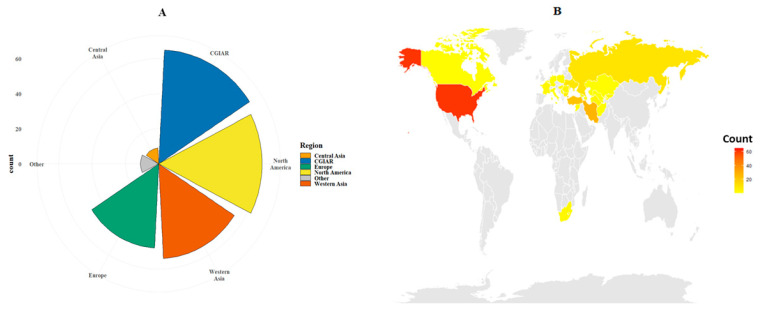
Characterization of the Winter Wheat Association Genetics Initiative (WWAGI) panel with 249 advanced lines and varieties used based on region and geographic origin (**A**,**B**).

**Table 1 plants-12-04053-t001:** Major allele frequency, sample size, genetic diversity, and the polymorphism information content (PIC) of Chromosome, Region of the Winter Wheat Association Genetics Initiative (WWAGI) with 249 advanced lines and varieties.

	Major Allele Frequency	Number of SNPs	Genetic Diversity	PIC
Chromosome				
1A	0.70	534	0.43	0.37
1B	0.70	710	0.42	0.36
1D	0.70	221	0.41	0.35
2A	0.73	661	0.38	0.33
2B	0.68	787	0.46	0.39
2D	0.68	234	0.43	0.36
3A	0.70	451	0.43	0.37
3B	0.67	777	0.47	0.40
3D	0.65	106	0.47	0.40
4A	0.72	383	0.4	0.35
4B	0.70	308	0.43	0.37
4D	0.67	32	0.44	0.36
5A	0.66	596	0.46	0.39
5B	0.72	815	0.41	0.36
5D	0.69	137	0.42	0.36
6A	0.69	503	0.44	0.38
6B	0.68	674	0.44	0.38
6D	0.72	76	0.4	0.35
7A	0.70	525	0.44	0.38
7B	0.68	470	0.45	0.38
7D	0.72	83	0.4	0.34
	Major Allele Frequency	Sample Size	Genetic Diversity	PIC
Region				
Central Asia	0.81	9	0.27	0.22
CGIAR	0.72	65	0.39	0.35
Europe	0.74	48	0.38	0.34
North America	0.72	62	0.39	0.35
Other	0.75	11	0.33	0.28
Western Asia	0.70	54	0.42	0.37

**Table 2 plants-12-04053-t002:** AMOVA of the Winter Wheat Association Genetics Initiative (WWAGI) with 249 advanced lines and varieties based on geographical region.

Region				
Source	df	Mean Sq	Sum Sq	Percentage
Among Region	5	28,330	141,652	6%
Within Central Asia	8	6843	54,748	2%
Within CGIAR	64	8921	570,931	26%
Within Europe	47	8685	408,211	18%
Within North America	61	8413	513,164	23%
Within Other	10	3830	38,298	2%
Within Western Asia	53	9118	483,267	22%

**Table 3 plants-12-04053-t003:** Population differentiation of the Winter Wheat Association Genetics Initiative (WWAGI) with 249 advanced lines and varieties calculated as the Nei genetic distance according to geographical region.

Region	Central Asia	CGIAR	Europe	North America	Other	Western Asia
Central Asia	0.00	0.05	0.07	0.08	0.10	0.06
CGIAR	0.05	0.00	0.03	0.03	0.06	0.01
Europe	0.07	0.03	0.00	0.04	0.06	0.03
North America	0.08	0.03	0.04	0.00	0.05	0.04
Other	0.10	0.06	0.06	0.05	0.00	0.06
Western Asia	0.06	0.01	0.03	0.04	0.06	0.00

**Table 4 plants-12-04053-t004:** Phenology experiment conditions during the three vernalization and photoperiod treatments.

Experiment	Vernalization (Days at 5 °C)	Photoperiod (Hour in Days/Night)
V+P+	49	20/4
V−P+	0	20/4
V+P−	49	14/10

**Table 5 plants-12-04053-t005:** Descriptive statistic of growing degree days (GDD) for the WWAGI used based on vernalization and photoperiod requirement experiment.

Treatment	Mean	Min	Max	Variance	Genetic Variance	Heritability
V+P+	1273.13	1013.9	1629.42	18,120.59	16,994.02	0.92
V−P+	1417.13	896.85	2241.87	138,772.3	124,085.4	0.91
V+P−	1705.05	1270	2150	35,540.15	25,163.1	0.81

**Table 6 plants-12-04053-t006:** ANOVA of growing degree days (GDD) for the WWAGI genotypes used based on vernalization and photoperiod requirement experiment (treatment).

	Df	Sum sq	Mean sq	F Value	Pr (>F)
Treatment	2	20,430,313	10,215,157	2681.48	<2.2 × 10^−16^ ***
Genotype	234	12,550,473	53,635	14.079	4.459 × 10^−16^ ***
GenotypeXTreatment	255	11,103,562	43,543	11.43	1.820 × 10^−14^ ***
Residuals	40	152,381	3810		

*** significant at the probability of 0.01%.

**Table 7 plants-12-04053-t007:** Significant marker associations for each trait under study.

Trait ^1^	Marker	Chromosome	Position	−log10(*p*)	R^2^
Strict photosensitive	*Ppd-D1 **	2D	36207497	11.82	0.41
Strict photosensitive	AX-94406957	2D	35069143	4.5	0.31
Strict photosensitive	AX-94656988	7D	197665753	3.46	0.25
Strict photosensitive	AX-94383404	7B	137485354	3.12	0.25
Strict vernalization	*Vrn-B1 **	5B	577015200	7.75	0.36
Strict vernalization	AX-95166397	5B	583535223	3.76	0.27
Strict vernalization	AX-95194633	1A	500679531	3.37	0.28
Strict vernalization	AX-94913041	5A	598190302	3.31	0.27
Strict vernalization	AX-94918769	5B	589762586	3.3	0.26
Eps	*Vrn-B1*	5B	577015200	3.74	0.12
RRP	AX-94523405	5D	398019882	3.68	0.1
RRP	AX-94669768	5A	377557215	3.04	0.09
RRV	AX-94796479	5A	590225939	5.09	0.49
RRV	*Vrn-A1*	5A	590399578	4.05	0.44
RRV	AX-94404219	5A	590702992	3.48	0.42
RRV	AX-94926419	5B	578719444	3.31	0.41
RRV	AX-94679692	2D	572933785	3.08	0.4

^1^ Relative response to vernalization (RRV), relative response to photoperiod (RRP), and earliness per se (Eps). * Significant marker that passes Bonferroni correction.

## Data Availability

The data presented in this study are available in the article. The raw MS files are available on request from the corresponding author.

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
