# Peer review of "Genetic Diversity and Genome-Wide Association Study for the Phenology Response of Winter Wheats of North America, Western Asia, and Europe"

_plants, 2023, doi:10.3390/plants12234053_

Round 1

Reviewer 1 Report

Comments and Suggestions for Authors

Review of Manuscript: Genetic diversity and genome-wide association study for the phenology response of winter wheats of North America, Western Asia and Europe

The manuscript describes the genotypic characterisation of a bread wheat panel of 249 entries of the winter wheat panel WWAGI, using the 35k Wheat Breeder’s genotype array and a few gene based markers (Vrn1, PpD1, ESP) population structure and genetic diversity were assessed using the genotype data. Furthermore, the entries were grown in climate chambers under different temperature and day length regimes and the phenotypic responses were recorded and discussed in the light of the genotyping results, specifically of the allelic series of the named genes.

The manuscript is well structured, clear and in general of good English. Unfortunately, the conducted analyses seem to be mainly done as a box-ticking exercise and many results are irrelevant or may have been predicted by checking the pedigrees of the lines.

The major problem with this project is, that the individual entries for this panel were specifically selected, so it was not a natural population. I also assume that the collection is not a representation of e.g. the growing areas, but that a specific selection was conducted to put

this panel together. Details on the reason for selecting the specific entries are not given in the materials and methods.

The assessment of population structure seems irrelevant, apart from being used in the GWAS, as this panel was put together by design and not as outcome of a natural evolution.

Results, like that the CIGAR bred lines are more diverse than other groups, are not surprising, as the specific aim of the CIGAR strategy is, to increase the genetic diversity by using germplasm from diverse sources. I suggest to move the parts on the population structure to the supplementary material.

The group sizes for a split by country are different, specifically the group of “Central Asia” and “Others” are considerably smaller that the other groups (8 and 10 respectively, in comparison to group sizes between 47 to 63). When doing statistical analysis, this has a strong potential of a bias in the result. Either the smaller groups could be excluded, or a comparison of groups of similar size, randomly selected from the larger groups, could be tested.

Similarly, comparing genetic diversity by AMOVA of only a few lines carrying a certain allele combination (e.g. Vrn-A1a/Vrn-B1a/vrnD1b, only 5 entries, 2% of the total number of entries) with one much larger group (vrn-A1b/vrn-B1b/vrnD1b) containing 154 entries is statistically nonsense.

Groups of equal size need to be compared in order to draw a conclusion. I suggest to remove the AMOVA for the allelic groups.

A list of the cultivars/entries forming the Winter Wheat Association Genetics Initiative panel is missing and should be presented in the supplementary material.

This manuscript is interesting as a resource paper, as other groups may want to employ a well characterised winter wheat panel for there research. In order to be able to do that, the genotype and phenotype data of the panel should be made publicly available and the seed for the panel should be made available via a seed/gene bank. Only then, the information gifen here will provide the community with the “valuable tools to drive crop improvement”, which the authors mention in their introduction.

The population structure in the discussion needs to be removed as it given here. A specifically assembled panel with deliberate subgroups is compared with panels which try to capture the distribution over regions. This is scientifically not sound.

Line 381: remove the word “useful” as all that can be said is that it is genetic diversity. If it is useful has to be shown.

Specific comments

Figure1 1 A: The manuscript fails to explain where the positions of the SNP markers come from. I assume they were determined by the lab that developed the 35k Wheat Breeders’ array. The information needs to be added and Figure 1 A should be moved to supplementary data as it is not an original result. I would also suggest to move Figure 1 B to the supplementary, as the linkage equilibrium is little discussed in the manuscript.

Description of the PCA, e.g. line 114-115: The results of the PCA method are not strictly determined regarding the positive or negative side of the axis. As such it is discouraged to talk about the ‘negative side’ of the PCA plot. It is better to describe the locations of possible clusters in regards of each other, e.g. by saying: “The first axis differentiated between wheat from Central Asia and European lines, which are located on opposite sides of the axis”. There are several further cases mentioning negative or positive sides, which should all be removed.

Line 182ff needs to be changed.

The phylogenetic tree does not represents the clustering of all entries through respective layers.

The geographic origins … are lined up against the phylogenetic trees (in three different layers). This shows an enrichment of geographic origin ....

Line 183: Vrn-1 and Ppd-1 (and not just Vrn and Ppd) alleles.

Line183: The phylogenetic can be interpreted to split the dataset into four clades.

There are many more splits in that tree, but the authors want to report on the major splits here.

Line200: Do you mean phenology experiment (Table 4) ? It says Table 1. What is the genotype effect? Does genotype means “entry” here? More consistency is needed.

Table4 lacks the description of treatment. What is WWP?

Table 5 lacks the description of treatment and genotype.

Figure 9: I don’t see the threshold for the Bonferroni correction.

Comments on the Quality of English Language

The quality of the English is good. Just minor occasional mistakes.

Reviewer 2 Report

Comments and Suggestions for Authors

This is a very nice manuscript with obvious utility for wheat breeders. Different vernalization, photoperiod and flowering time loci were analyzed in this study. Figures 5 through 7 describe different frequencies of these alleles when planted world-wide. I can conjecture why some of the varieties are chosen in North America and Central Asia, but don’t know enough about the constraints on winter wheat in other areas of the world. I request that the authors explain why certain Eps, PPD and VRN genotypes are more common in some each of the regions.

Minor issues or questions:

How was the number of growing days determined in this study?

When referencing experiment, the names of the research group are needed, as well as the citation in brackets.

In table 1, is “sample of SNPs,” the number of SNP markers used on this chromosome?

Formatting for “Region” and “Chromosome” is off in Table 1

In Table 5, should the heading for the third line be GenotypeXTreatment?

In figure 9, add line for log 3 for Eps, RRP and RRV
